# Annotation and Identification of Phytochemicals from *Eleusine indica* Using High-Performance Liquid Chromatography Tandem Mass Spectrometry: Databases-Driven Approach

**DOI:** 10.3390/molecules28073111

**Published:** 2023-03-30

**Authors:** Nur Syahirah Mad Sukor, Zikry Hamizan Md Zakri, Nurulfazlina Edayah Rasol, Fatimah Salim

**Affiliations:** 1Faculty of Applied Sciences, Universiti Teknologi MARA (UiTM), Shah Alam 40450, Selangor, Malaysia; 2Institute for Tropical Biology and Conservation, Universiti Malaysia Sabah, Jalan UMS, Kota Kinabalu 88400, Sabah, Malaysia; 3Atta-Ur-Rahman Institute for Natural Product Discovery (AuRIns), Universiti Teknologi MARA Selangor Branch, Puncak Alam Campus, Puncak Alam 42300, Selangor, Malaysia

**Keywords:** *Eleusine indica*, LCMS, GNPS, MZmine, Compound Discoverer, SIRIUS

## Abstract

*Eleusine indica* (L.) Gaertn is a perennial herb belonging to the Poaceae family. As the only species of *Eleusine* found abundantly in Malaysia, it is locally known as “rumput sambau” and has been traditionally used to treat various ailments including pain relief from vaginal bleeding, hastening the placenta delivery after childbirth, asthma, hemorrhoids, urinary infection, fever, and as a tonic for flu-related symptoms. A diverse array of biological activities have been reported for the plant, such as antimicrobial, cytotoxic, anticonvulsant, anti-inflammatory, analgesic, antipyretic, and hepatoprotective action. Despite many reports on its traditional uses and biological activities, limited chemical databases are available for the plant. Thus, the aims of this study were to annotate and identify the phytochemical constituents in the methanolic extract of *E. indica* through tandem LCMS-based analysis techniques using MZmine, GNPS, Compound Discoverer, and SIRIUS platforms. This technique managed to identify a total of 65 phytochemicals in the extract, comprising primary and secondary metabolites, and was verified by the isolation of one of the identified phytochemicals. The structural elucidation mainly using 1D and 2D NMR as well as comparison with values in the literature confirms the isolated phytochemical to be a 3-OH anomer of loliolide, a benzofuran-type of compound, which consequently increases the level of confidence in the applied technique. The research describes a useful method for the fast and simultaneous identification of phytochemicals in *E. indica*, contributing to the study of the chemical properties of the genus and family.

## 1. Introduction

Identification of phytochemicals is crucial in the investigation of plant samples. In the past two decades, new technologies and methods for structural identification have come forth, which promote the speed and accuracy of phytochemical analysis [1]. The liquid chromatography coupled to mass spectrometry (LCMS) approach in structural identification has gained popularity due to its high throughput, soft ionization, and good coverage of phytochemicals [2]. LCMS is the best approach in plant chemicals analysis due to its versatility, sensitivity, and ability to separate and detect highly diverse semi-polar compounds, including key secondary metabolite groups. Tandem MS analysis is important to acquire both precursor and fragment ion information which can be used to annotate, identify, and dereplicate phytochemicals by providing a wealth of precise structural information [3]. LCMS-based phytochemical analysis can be classified into two types, namely, untargeted and targeted approaches. The former refers to a comprehensive analysis of all the measurable chemicals including the unknowns, while the latter focuses on the measurement of defined groups of chemicals [4]. The analysis can be facilitated with metabolite annotation tools such as MZmine, Global Natural Social Molecular Networking (GNPS), Compound Discoverer, and SIRIUS 4.0 [5]. This approach leads to the structural characterization of phytochemical mixtures, especially through identifying biomarkers and minor components which consequently facilitate and accelerate the discovery of novel active compounds [6].

*Eleusine indica* (L.) Gaertn is a perennial herb belonging to the Poaceae family that has been utilized widely for its medicinal values. The plant is widely spread in tropical regions and most of the Pacific Islands [7]. In Malaysia, *E. indica* is locally known as “rumput sambau” and it is the only species of *Eleusine* that can be easily found; it grows abundantly as a weed along roads and pavements [8]. *E. indica* has been used as traditional medicine around the world to treat various ailments including symptoms related to microbial infection, sprained muscle, coughing blood, and centipede or scorpion poisoning [9]. In Peninsular Malaysia, the plant’s leaves are pounded to extract the juice, which is used to hasten the delivery of placenta for women after childbirth and to relieve pain during vaginal bleeding. The root decoction is used in treating asthma, while the decoction of the whole plant is used to treat urinary infections [10,11]. In East Malaysia, Kadazandusun people used an infusion of the plant’s aerial part with rice to treat symptoms related to flu viral infection, and the decoction of roots mixed with *Capsicum* sp. (Solanacae) to treat piles [9,12]. This plant has a diverse array of biological activities including antioxidant, antibacterial, cytotoxic, anticonvulsant, anti-inflammatory, antidiabetic, antiplasmodial, hepatoprotective, analgesic, antipyretic, and others [7,13,14]. Hitherto, only a few phytochemicals have been isolated from the plant; they include schaftoside, vitexin, and isovitexin, β-sitosterol, stigmasterol, 3-O-β-D-glucopyranosyl-β-sitosterol and its 6′-O-palmitoyl derivatives, 1-[[[(2-aminoethoxy) hydroxyphosphinyl]oxy]methyl]-1,2-ethanediyl ester, and hexadecanoic acid [15,16,17,18,19]. An LCMS metabolite profiling and fingerprinting on the plant extract has identified p-coumaric acid and isoschaftoside along with a series of primary metabolites and amino acids [20].

Despite many reports on the traditional uses and biological activities of *E. indica*, not many phytochemicals have been isolated, thus limiting the plant’s available chemical databases. Therefore, the present study was undertaken to explore the chemistry of *E. indica* through tandem LCMS-based analysis. This paper discusses the analysis process for the characterization and identification of phytochemicals in the methanolic extract of the plant employing annotation platforms of MZmine, GNPS, Compound Discoverer 3.0 and SIRIUS; the platforms were integrated with several available spectral and compound databases as well as a custom-built one based on the Poaceae family. In addition, the isolation and structural elucidation of an identified phytochemical are also reported here, to verify the reliability and to increase the level of confidence in the technique. This research describes a useful method for the fast and simultaneous characterization and identification of phytochemicals in *E. indica*, contributing to knowledge about its genus and family chemical properties.

## 2. Results and Discussion

Figure 1 shows an infographic summarizing the steps of the research: the methanolic extraction, tandem LCMS and data analyses; the use of MZmine, Compound Discoverer (CD), GNPS, and SIRIUS platforms for annotation and identification; the use of a Venn diagram to display the distribution of the annotated phytochemicals in each platform; and the verification of the technique through the isolation and structural elucidation of phytochemical **1**. The detail for each step is explained in the subsection below.

### 2.1. Tandem LCMS Analysis for Phytochemicals Annotation and Identification

In this study, a comprehensive high-resolution MS in a data-dependent full-scan acquisition method was developed to separate and detect the phytochemicals in *E. indica* methanolic extract. The phytochemicals profile was composed of hundreds of features that are recognized by their measured mass-to-charge ratio (*m/z*), retention time (Rt), and relative abundance. The annotation of the phytochemicals was performed using a library of natural products that contains the previously reported phytochemicals from the plant family, Poaceae. The library was custom-built by plant names (with all synonyms) that were queried in the Dictionary of Natural Products (DNP) Ver. 26.2 (December 2017), and all resulting hits were used to build a library of natural products. All the detected phytochemicals were screened against the prepared library using MZmine 2.53, Compound Discoverer 3.0, GNPS, and SIRIUS platforms; this enabled comparison of the mass errors (ppm) and isotopic patterns of the phytochemicals in the library with the observed mass spectra and ranking of the probable identity of the phytochemicals based on match score. The combination of these annotation tools accelerates the process of identification of the phytochemicals. This approach has been shown to succeed in the identification of phytochemicals in a number of studies including fully clarifying the chemical constituents of a herbal medicine in China, the Pingxiao capsule (PXC) [21]. The processes for annotation and identification of the phytochemicals in *E. indica* methanolic extract through the mentioned platforms are discussed in detail below.

#### 2.1.1. Data Processing, Enrichment and Phytochemicals Annotation by MZmine 2.53

The total ion chromatogram (TIC) of *E. indica*’s phytochemicals profile (Figure 2) acquired from the optimized 30-min gradient elution gave *m/z* features in a range of 166.0859–806.5875. Pre-processing of the positive ion mode raw data file using MZmine resulted in 426 *m/z* features. Annotation of these features was carried out based on accurate mass (MS^1^) information with the curated DNP and custom databases (as previously outlined). To characterize the phytochemicals, a specialized compound database was retrieved from the online DNP database. The biological source keyword, Poaceae (family) yielded 1106 compounds. To supplement this database, a custom library generated from the previously isolated compounds of *E. indica* was added [14,15,20]. Both databases were imported to MZmine and employed as the custom-built database for peak identification. Hits were manually cross-checked against the MS/MS spectral fragmentation data. In order to ensure that none of the peaks overlapped, a 3D chromatogram plot (front, left and right view) was generated for the profile. As there were many overlapping peaks at the same Rt, the highest *m/z* abundance that represented a specific Rt was selected.

As shown in Table 1, the above processing of the extract managed to annotate 12 phytochemicals based on the custom-built database (LR and DNP sourced). Three of the phytochemicals were detected at several different retention times, namely, loliolide (**1**, **1′**), isoschaftoside (**2**, **2′**), and adenosine (**5**, **5′**); they were assigned the same acronym but with the addition of prime (‘) to the duplicate to ease confusion. This redundancy in detection is probably due to the stereoisomerism of the compounds, which may affect their polarity [22]. Notably, MS spectra typically cannot differentiate the stereoisomers, and additional experiments, including comparison with standards, are required to assign the absolute structure [23]. The annotated phytochemicals, loliolide (**1**, **1′**), isoschaftoside (**2**, **2′**), and vitexin (**3**) were hit with the custom database developed through the LR, and 4-ethoxy-6-methoxy-2-(8,11,14-pentadecatrienyl)-1,3-benzenediol,5-Ethoxy-3-(10,13,16-heptadecatrienyl)-1,2,4-benzenetriol (**4**), oryzamutaic acid E (**7**), 1-feruloyl-2-hydroxyputrescine (**8**) and 2-[2-(3-Methoxyphenyl) ethenyl]-4H-3,1-benzoxazin-4-one, 2-(3-methoxycinnamoyl)-4H-3,1-benzoxazin-4-one (**9**) were detected using the enriched database from DNP. Adenosine (**5**, **5′**) was hit by both LR and DNP sourced. Of these annotated phytochemicals, only **2** and **3** have been previously reported as the constituents of *E. indica* [14,20,24]. Since the LR database was only focusing on secondary metabolites, it seems that the DNP database complemented it by annotating some common compounds and primary metabolites from the plant extract. All the phytochemicals annotated in MZmine were identified with confidence level 3 due to only MS^1^ data characterization [25,26]. To acquire a higher level of confidence, the structures must be further characterized with their MS^n^ using GNPS, CD and SIRIUS platforms.

#### 2.1.2. Phytochemicals Annotation by GNPS

The same LCMS pre-treated data by MZmine, MGF file and feature table were used and uploaded into the Feature-Based Molecular Networking (FBMN) workflow, in GNPS. FBMN is an advanced method for molecular networking that provides accurate ion abundance for statistical analysis and support for isomer resolution or ion mobility. It has been validated to be an effective strategy for the identification of natural compounds from different sources [27,28]. GNPS facilitated the phytochemicals annotation through the comparison of the spectra from experimental data with open-access reference spectral libraries [28]. In GNPS, the annotation is achieved through the cosine value which refers to a normalized dot-product, a mathematical measure of spectral similarity between two fragmentation spectra and their library class that determines the level awarded, whether gold, silver or bronze. A cosine score of 1 represents identical spectra while a cosine score of 0 denotes no similarity at all. The degree of confidence in the annotated phytochemicals is determined through the cosine value and library class index used globally for each cluster across all networking views [28].

As shown in Figure 3, analysis through the GNPS platform managed to annotate 14 phytochemicals, of which three (**1**, **2** and **3**) were also hit in the MZmine platform. The detailed features of the 14 annotated phytochemicals are listed in Table 2, which includes their library class, cosine, spectral and library *m/z*, ionization used, instrumentation, and ion source. With all the library hits, the mirror match between the experimental and library mass spectra can be obtained. For example, the mirror spectral match for isoschaftoside (**2**) shows a very close similarity (gold level) to the experimental data (Figure 4), with a cosine value of 0.82. This confirms that the *m/z* 565.26 peak detected in the methanolic extract can be putatively annotated as isoschaftoside. An *m/z* peak at 433.11 with a cosine value of 0.96 was annotated as vitexin (**3**) similar to that annotated in MZmine platform. This match was also classified in the GNPS platform as a gold level due to the excellent quality of a match with the experimental spectrum considering the mass accuracy of the reference spectrum (resolution and calibration of the instrument), sample type, experimental setup, and associated sample information (metadata). It is worth mentioning that annotations **2** and **3** were matched with the Bioinformatics and Molecular Design Research Center Mass Spectral Library—Natural Products (BMDMS-NP), which contains high reliability references on experimental spectral data of metabolites. In fact, BMDMS-NP reports that the reference data for the two phytochemicals were obtained from the same instrument used in the present study (orbitrap), thus further illustrating the reliability of the result. Another phytochemical that shared high reliability in the annotation is loliolide (**1**) of cosine value 0.94 and bronze match classification. Isoschaftoside and vitexin have been reported previously as constituents of this plant [14,20].

#### 2.1.3. Phytochemicals Annotation by Compound Discoverer (CD) 3.0

The pre-treated LCMS data was further analyzed through—CD platform for MS/MS fragmentation patterns from various spectra databases. Among the applied databases were mzCloud spectral library, mzVault, ChemSpider™, Human Metabolome Database (HMDB), Kyoto Encyclopaedia of Genes and Genomes (KEGG), Massbank, Biocyc, NIST and Drugbank as well as our own custom-built database from the Poaceae family. Here, exact mass and the chemical formula were calculated where several compounds with similar masses and formulas were selected as candidates. To ensure a better hit, entries with higher ppm errors (>10 ppm) were discarded from this analysis. However, in the present sample, it was observed that mass errors were below 2 ppm in most cases. The fragment ions in the MS/MS data were analyzed in silico; the results were generated by manually dissecting the molecules at various possible sites and comparing the theoretical fragments with those obtained from the data.

Table 3 shows the 38 phytochemicals along with their features (Rt, molecular formula, mzCloud similarity, FISh scoring and MS information) that could be annotated using CD software. FISh scoring provides fragment assignment based on mzCloud literature and in silico fragmentation rules (MS^2^ and MS^3^ data), and comparisons to the parent molecule for phytochemical assignment. The identification was either formal (when at least two physicochemical parameters, such as chromatographic retention time and MS/MS spectrum, matched those of our spectral library of reference compounds) or putative (based on information from mzCloud and the interpretation of MS and MS/MS spectra), corresponding to levels 1 and 2 from the metabolomics standard initiative [29,30]. In the present work, the molecular formula of the phytochemicals was mainly considered to be putative based on the predicted composition on the platform.

For consistency, here again, the phytochemical annotated as isoschaftoside (**2**) will be used to discuss the CD features. As shown in Table 3, compound **2** was annotated with good accuracy by exhibiting only 0.00113 Da (∆Mass) and 2 ppm (∆Mass) mass error; the mzCloud similarity match was 97.1% and the FISh score was 45.16; it was annotated as level 2 [31]. Figure 5 shows the presentation of how the in silico spectral fragmentation in the mzCloud library of **2** matches the *m/z* peak 565.26 detected in the plant extract. The number of (green) fragments ions indicate that majority of the signals in the spectrum is matched with the mzCloud library, consequently increasing the confidence in identification. This compound also matched with the custom database from Poaceae family and has been isolated, elucidated and characterized previously from *E. indica* [14,20].

Another annotated phytochemical in the extract that worth highlighting is adenosine (**5**). The in silico fragmentation from the FISh algorithm structurally explained more than 83.33% of the fragment ions for adenosine (Table 3). Adenosine is an organic compound that occurs widely in nature in various derivatives. It participates in improving the action of the plant on memory impairment and increased cyclic adenosine monophosphate (AMP). This compound was previously isolated and identified from *Anredera cordifolia* [32]. However, this is the first report on adenosine in *E. indica*. Figure 6 shows the TIC containing the structures of all the phytochemicals annotated by the CD platform with their respective Rt. Although the in silico fragmentation behavior in the CD platform provides an efficient characterization feature that will accelerate the annotation process of the phytochemicals, this data still needs to be confirmed with the MS/MS spectra. Thus, SIRIUS platform was further applied in the analysis to confirm the phytochemicals’ fragmentation pattern.

#### 2.1.4. Phytochemicals Annotation by SIRIUS 4.0

SIRIUS is a software dedicated to the annotation of ions from fragmentation spectra. This software complements the other platforms described above, particularly the CD platform. First, SIRIUS computes the candidate molecular formula (MF) by (1) matching the MS^1^ experimental spectra against the predicted isotopic pattern and (2) establishing how much the fragmentation spectra can be explained by the candidate MFs using fragmentation trees. SIRIUS integrates other algorithms or models such as: ZODIAC for improved MF prediction; CSI: FingerID for putative annotation of structure and COSMIC for establishing confidence in the match; or CANOPUS for putative chemical class annotation. SIRIUS has an advanced graphical user interface, and the tools can be run in command line mode. This work presents the COSMIC (Confidence of Small Molecule IdentifiCations) workflow that combines the selection or generation of a structure database, searching in the structure database with CSI: FingerID and a confidence score to differentiate between correct and incorrect annotations. Candidate structures and database-independent fingerprint vectors were obtained by loading the above-mentioned MGF files into the SIRIUS and CSI-FingerID pipeline. Data were acquired in a positive mode due to the higher sensitivity and the higher quality of fingerprint predictions of SIRIUS + CSI-FingerID compared to the negative mode. After computing the processed data (MS^2^) into this platform, the results of each feature are displayed through the Rt, *m/z* and COSMIC value data.

A fragmentation tree annotates peaks in the fragmentation spectrum with molecular formulas and identities of likely losses between the fragments, similar to “fragmentation diagrams” created by experts. For each fragmentation spectrum, COSMIC considers only the structure candidate that is top ranked by CSI: FingerID as an annotation; COSMIC neither changes annotations (re-ranks structure candidates) nor discards any annotations. COSMIC’s confidence score combines E-value estimation and a linear support vector machine (SVM) with enforced directionality. The calculated tree must not be understood as ground truth but can be used to derive information about the measured phytochemical. The fragmentation tree is computed from the fragmentation spectrum given the (candidate) molecular formula of the precursor ion. Initially, a fragmentation graph is constructed in the following way. For every fragment peak, all possible molecular formula explanations are computed. These explanations must be subbing formulas of the precursor molecular formula fragment that only loses atoms, and never gains new atoms. Every such molecular formula is a node in the graph. Nodes are connected by an edge if one node is a subformula of another representing a potential loss. Using combinatorial optimization, the best scoring fragmentation tree is computed which explains every peak at most once. Unexplained peaks are considered noise. Figure 7 shows the example of a fragmentation tree for candidate **2** which is annotated as isoschaftoside. As shown from the fragmentation tree, the *m/z* 565.1557 C_28_H_28_0_14_ loses H_2_O (18 Da) and then produces a molecular ion 547.1456 C_28_H_28_0_13_. From the molecular ion 547.1456, it can be observed that the structure could be fragmented into two molecular ions: [M + H-H_2_O] with molecular formula C_28_H_24_0_12_ at *m/z* 529.1347 and [M + H-CH_4_O_2_] with molecular formula C_25_H_22_O_11_ at *m/z* 499.1245. This can guide the understanding on how the fragmentation pathway occurred during the analysis.

Candidate structures for the phytochemicals of *E. indica* fed and non-fed pitchers were obtained by searching the top hit of CSI-FingerID in all databases and manually curating the results; for all the other analyses, fingerprint vectors of the top 10 candidates of all predicted formulas were exported. Annotation candidates were sorted by their score and the similarity between the predicted fingerprint and the fingerprint of each candidate. The higher the percentage, the higher the similarity. Candidates can be filtered by database, SMARTS string and XlogP value. In the SIRIUS platform, the practical databases used for annotation of phytochemicals would be Natural Products, COCONUT, and CHEBI, which are databases for natural products. It is worth mentioning that the unique feature of the SIRIUS platform is the CANOPUS which can predict the classification (i.e., functional group) of the phytochemicals. This would help to narrow down the search for the correct phytochemical annotation. Notably, COSMIC complements compound class annotation tools such as CANOPUS. COSMIC targets molecular structure annotations but annotates only a fraction of the compounds in a sample; in contrast, CANOPUS annotates practically all compounds in a sample for which fragmentation spectra have been measured but is restricted to annotating compound classes. Hence, both methods provide viable information. Which method is better suited depends on the underlying and focus of the research [33]. For example, COSMIC classified a phytochemical with an Rt of 5.00 min, and *m/z* peak of 565.16 to be from a kingdom of organic compounds, superclass of phenylpropanoids, class of flavonoids and subclass of flavonoid glycosides. The structure was further narrowed into a flavonoid C-glycosides compound type and flavonoid 8-C-glycosides. From this information and the literature values, the phytochemical that possibly satisfies these features is isoschaftoside (**2**). In fact, the MZmine, CD and GNPS platforms also support this annotation. This is evidence that COSMIC features can accelerate the selection of the candidates to be annotated. In addition, the level of confidence is evaluated through the metabolite identification confidence (MIC) level which is ranked 1–5 for putative metabolite annotation. Table 4 displays 20 phytochemicals that managed to be identified through the SIRIUS platform along with their MIC values. Table 5 lists 42 phytochemicals that could be characterized by their respective classes of compounds.

### 2.2. Verification of the Identified Phytochemicals

The platforms used in the present work each have their own strengths in the phytochemical annotation and identification process. The Venn diagram in Figure 8 below shows the performance of each platform in annotating the phytochemicals of *E. indica*. It can be clearly seen that only three phytochemicals, namely, loliolide (**1**), isoschaftoside (**2**), and vitexin (**3**), could be consistently identified through the four platforms. The CD platform gave the highest number of annotated phytochemicals (40) which is probably due to the large databases integrated into the platform, as mentioned earlier. As expected, the MZmine platform gave the lowest number of annotated phytochemicals (7) since it only integrates with a custom database built based on the phytochemicals previously reported from the Poaceae family. Since each platform gave a diversity of phytochemicals, it is suggested to use all of the platforms on a complementary basis, to annotate and identify different phytochemicals.

Even though the present work has managed to annotate and identify 65 phytochemicals from the methanolic extract of *E. indica*, some of the previously isolated constituents were not detected in the extract. These annotated and identified phytochemicals will remain tentative if no reference standard was used or the method is not verified. Thus, in an attempt to assure the reliability of the LCMS analysis in annotating phytochemicals in *E. indica* methanolic extract, one of the consistently identified phytochemicals on all the platforms, loliolide (**1**), has been subjected to a further isolation and purification process. From the total ion chromatogram of LCMS, compound **1** was predicted to elute at minute 6.78 (Table 1, Table 3 and Table 4). In order to isolate and purify **1**, the extract was subjected to chromatographic monitoring techniques including semi-prep HPLC, and recycling HPLC yielded 10 mg of **1**. The structure of phytochemical **1** was elucidated through various spectroscopic techniques and comparisons with the literature values. The physical and spectroscopic characterization of phytochemical **1** was as follows.

Phytochemical **1** was isolated as a colourless crystal possessing a melting point of 167 °C. Its MS spectrum showed a molecular ion peak at *m/z* 197 consistent with the molecular formula of C_16_H_11_O_3_. The UV spectrum acquired in MeOH displayed absorptions at λ_max_ 208, 265 and 349 nm. The IR spectrum showed absorption bands characteristic for a benzofuran type of compound at 3433 cm^−1^ for a hydroxy group, while strong stretches at 1731 and 1024 cm^−1^ indicated the presence of a conjugated cyclic ester carbonyl group (C=O), and an ether group (C-O), respectively [34].

The ^1^H NMR spectrum of **1** displayed a singlet at δ_H_ 5.69 which corresponds to H-7 due to the presence of one double bond. The resonances in the medium region at δ_H_ 4.33 (m, 1H) indicated the neighboring hydrogen at H-3. Two pairs of methylene can be observed at δ_H_ 2.01 (*dt*, J = 14.4, 3.1 Hz, 1H) and 1.55 (*dd*, J = 14.4, 3.7 Hz, 1H) and 2.46 (*dt*, J = 13.6, 2.4 Hz, 1H) and 1.78 (*d*, J = 4.0 Hz, 1H). The splitting pattern of these resonances is ascribable due to the presence of cyclic methylene. Two methyl resonances were observed close to each other at δ 1.27(*s*, 3H) and 1.47 (*s*, 3H) which indicated the presence of geminal dimethyl while another methyl was deshielded at δ1.78 (*s*, 3H) because of the presence of neighboring oxygen atom. The stereochemistry of C-3 and C-5 was determined based on the NOESY experiment and chemical correlation method. The NOESY correlation of δ_H_ 4.33 (H-3) with δ_H_ 1.78 (H_3_-11), 1.55 (H-2β), and δ_H_ 2.44 (H-4β) indicated that they are on the same phase, subsequently established H-3 to be β-oriented. This consequently assigned the 3-OH group to be α-positioned. The assignment is further confirmed by a comparison of the chemical shifts observed for the H-3β and H-3α protons in the H-3 anomer of loliolide [35]. The H-3α proton appeared at a slightly lower field at δ_H_ 4.84 as it was spatially away from the H-11 methyl electron cloud.

A total of 11 carbon resonances were demonstrated from the ^13^C-APT NMR spectrum, of which one conjugated ketone carbonyl resonance was observed at δ_c_ 181.5. Other than that, quaternary carbon resonances were observed including a deshielded trisubstituted methine carbon at δ_c_ 170.1 which belongs to C-6, while C-5 shifted downfield due to neighboring oxygen. Moreover, comparative analyses of the ^13^C-APT and ^1^H NMR spectrum showed the resonances of three methyl groups at δ_c_ 25.2, 26.5 and 29.7, a trisubstituted olefinic bond that appeared at δ_c_ 112.2 and 170.1, and a secondary hydroxy group at δ_c_ 66.1 ppm. The ^1^H and ^13^C-APT NMR data suggested that **1** is a bicyclic molecule, which led to the benzofuran type of compound. Comparison with literature data [35,36] confirmed that **1** is a benzofuran type of compound known to be loliolide, a 3-OH β-oriented structure. However, based on the deviation in the chemical shift of protons and carbons at positions 2, 3 and 4 from the reported data (Table 6), the stereochemistry at position C-3 is believed to be different. Based on the NOE correlations discussed earlier, the assignment of 3-OH was established to be α-oriented, consequently elucidating compound **1** as the C-3 anomer of loliolide (Figure 9). Loliolide was previously reported as a constituent of other plants including *L. salicaria, H. angiospermum, A. lappa, S. oleraceus, P. campanulatus* (Cav.), *P. indicus, M. alba,* and *M. whitei* [37]. The isolation and structural elucidation of **1** as loliolide verified that structural annotation using tandem LCMS analysis is reliable. However, the elucidation also supports the limitation of MS spectra in differentiating stereoisomers.

## 3. Materials and Methods

### 3.1. Materials and Reagents

#### 3.1.1. Plant Materials and Extraction

*E. indica* was collected in Tanjung Karang, Malaysia. The plant specimen was identified by a certified botanist, En. Ahmad Zainudin Ibrahim, and a voucher specimen with the code DBKL177 was deposited at the Herbarium Taman Botani Perdana Kuala Lumpur. The aerial and leaf parts of the plant (10 kg) were cut into small pieces and dried in an oven at 40 °C. The dried sample was weighed and ground before being extracted using methanol at room temperature for 72 h. The extract was filtered, and the solvent was evaporated under reduced pressure, resulting in 41.62 g of methanol extract. The extract was stored at 4 °C before analysis.

#### 3.1.2. Chemicals and Solvents

All AR (analytical reagent) grade chemicals used in this study were purchased from reputed manufacturers. Methanol (MeOH), and acetone (Ace) were of analytical grade. MeOH and acetonitrile (MeCN) HPLC grade were purchased from RCI Labscan (Bangkok, Thailand) and ultra-pure water (UPW) was from Sartorius.

### 3.2. LCMS/MS Condition

#### 3.2.1. Sample Preparation

Solid phase extraction (SPE) using Strata^®^ C18E (55 µm, 70 A), Phenomenex cartridges (500 mg, 6 mL) was employed for sample clean-up and pre-concentration. To activate the cartridges, 6 mL UPW was used, followed by 6 mL methanol. Before loading a 2 mL crude extract, the cartridge was equilibrated with 6 mL of 95% MeOH at a constant flow rate. Elution was performed with 5 mL of 95% MeOH. The extract was dried using a vacuum concentrator [38]. Then, 2 mg of the extract was dissolved in MeOH and filtered through 0.22 μm syringe filters into a vial, capped, and submitted for LCMS analysis.

#### 3.2.2. LCMS Optimization

LCMS analysis was performed using a Phenomenex reversed-phase Kinetex XB-C18 column (100 × 2.1 mm, 100 Å, 1.7 μm particle size). Mobile phase A was UPW and mobile phase B was LCMS grade MeCN. A constant flow rate of 0.8 mL/min was used, and the mobile phase gradient was: 0 min; 10% B, 20 min; 100% B, 30 min; 100% B. The column was equilibrated with 90% mobile phase A for 15 min before the next injection. The column oven was set at 35 °C, and the full loop injection volume was set at 5 µL [39]. The LCMS instrument used was a Thermo Scientific Orbitrap Elite with electrospray ionization (ESI) in positive mode. The resolving power for accurate mass measurement during the LCMS run was 120 K defined at *m/z* 400. The instrument was externally calibrated with Thermo Pierce calibration solution before LCMS runs. Full scan mode was used to record all the masses in the range of 100–600 *m/z*. In addition to the full scan, data-dependent MS/MS fragmentation was recorded for the 5 tallest peaks on each spectral scan with various collision energies. The spectrum data obtained from the LCMS analysis were processed using several available platforms such as MZmine, GNPS, Compound Discoverer, and SIRIUS.

### 3.3. LCMS Data Analysis

#### 3.3.1. MZmine 2.53 Data Pre-Treatment and Processing

The raw data were converted into mzML files using ProteoWizard function msconvert and were then processed using MZmine software version 2.53 [40] with the following steps: peak detection, chromatograph builder, chromatogram deconvolution, deisotoping, peak alignment, duplicate peak filtering, peak list row filtering, and gap-filling. The parameters for these steps were adjusted based on the centroid mass detector, noise level, minimum group size of scans, minimum group intensity threshold, *m/z* tolerance, S/N estimator and ratio, minimum feature height, coefficient area threshold, peak duration ranges, Rt wavelet range, isotopes peaks grouper, adduct search, complex search, *m/z* tolerance for peak alignment, absolute Rt tolerance, weight for *m/z*, weight for Rt, and peak-list rows filter. The online DNP database was used to create the custom compound database [41]. The resulting MS^1^ feature data were exported to excel (.csv) and the MS^2^ feature data were exported to SIRIUS and GNPS as an MGF file.

#### 3.3.2. Global Natural Social Molecular Networking (GNPS)

After processing the LCMS/MS data with MZmine software, the data were exported in two formats (TXT or CSV): a table containing the intensities of LCMS ion features, and an MS/MS spectral summary file in MGF format that contained a list of MS/MS spectra associated with the LCMS ion features. These files were then used as input for the SuperQuick FBMN tool, which was accessed using GNPS credentials and email. The “Feature Generation tool” was selected, and the feature quantification table and MS/MS spectral file were uploaded to the tool. After clicking on “Analyze Uploaded Files with GNPS Molecular Networking”, the FBMN job was completed, and the results were inspected on GNPS. Once the job was finished, an email notification with a link to the results page was sent, which could take any time from 10 to 10 h depending on the number of samples and the instrument used [28].

#### 3.3.3. Compound Discoverer™ 3.0

Data processing was carried out through Natural Product Unknown ID with Online and Local Database Searches method. This method involves several steps to detect and identify unknown phytochemicals, including Rt alignment, unknown phytochemical detection, and phytochemical grouping across all samples without statistics. In addition, elemental compositions were predicted for all phytochemicals, and the chemical background was hidden using blank samples. The phytochemicals were then identified using various tools, including mzCloud [data dependent MS^2^ (ddMS^2^) and/or data-independent acquisition (DIA)], ChemSpider (exact mass or formula), and local database searches against Mass Lists (exact mass with or without Rt) and mzVault spectral libraries. A spectral similarity search was performed against mzCloud for compounds with ddMS^2^, and mzLogic was applied to rank order structure candidates from ChemSpider and mass list matches. Finally, spectral distance scoring was applied to ChemSpider and mass list matches [42].

#### 3.3.4. SIRIUS 4.0

The MGF file from MZmine was imported to the SIRIUS software where CSI-FingerID pipeline was applied to further annotate the phytochemicals in the plant. The data extraction (MS/MS) was carried out using an in-house built code that searched for fragmentation events triggered in a window of 0.5 min within the feature Rt. To avoid misidentification of closely eluting isobaric phytochemicals, the maximum intensity in the MS^1^ extracted ion chromatogram (XIC) of the feature *m/z* (with 5 ppm error) that was closest to the feature Rt was searched. After determining the Rt window for a selected feature, all fragmentation events (MS/MS data) whose parent ions matched the feature *m/z* ratio within a 5 ppm error were stored within the new Rt window. CSI-FingerID was used to generate candidate structures for the phytochemicals of *E. indica* fed and non-fed pitchers. The top hit from this search was manually curated to ensure accuracy. For all other analyses, fingerprint vectors of the top 10 candidates of all predicted formulas were exported. When multiple adducts were present in a feature, only the formulas that matched the adducts’ formulas were retained. Then, only fingerprints that explained over three peaks and over a third of the intensity were retained. The final selection of fingerprint vectors was determined by collapsing all adducts per feature and retaining only those fingerprint vectors corresponding to the candidate with the highest score and those with a percentage of less than 30%. If the candidates had a greater COSMIC value and CSI: FingerID matching percentage, their fingerprints were retained. The following SIRIUS molecular formula calculation parameters were created to proceed with the analysis of the benchmarking dataset: potential ionisation, [M + H]^+^; instrument, orbitrap; tolerance, 50 ppm; candidate molecular formulas, 3; filtered by formulas from biological databases. For the CSI: FingerID process, the following parameters were used: potential adducts, [M + H]^+^; filter, compounds present in the biological database; maximum number of returned candidates, infinite [43,44].

### 3.4. Isolation and Purification of Compound ***1***

#### 3.4.1. General Chromatographic Procedure

The semi-preparative HPLC analysis was carried out using a DIONEX Ultimate 3000 HPLC system from ThermoFisher, Waltham, MA, USA. The system included a photodiode array detector (PDA), an auto-sampler injector, a fraction collector, and a 10 mL sample loop. The separation was performed on a Hypersil GOLD C18 column from Thermo Scientific with a pore size of 175 Å and dimensions of 250 mm × 10 mm (5 μm particle size). The instrument was controlled by software Chromeleon version 7.2 provided by the supplier, and data analysis was also conducted using this software. Recycling HPLC was performed on a JAI model LC-9103 from Japan Analytical Industry Co., Ltd. (Mizuho, Tokyo, Japan) equipped with a reciprocating double plunger pump type P-9140B and a UV detector with a wavelength set to 210 nm. The separation was carried out on preparative columns JAIGEL-ODS-AP, SP-120-15 and GL Sciences- Inertsustain Column C18, both with dimensions of 20 mm × 250 mm (10 µm particle size).

#### 3.4.2. Semi-Preparative and Recycling HPLC

The extract was weighed precisely at a concentration of 8 mg/mL and dissolved in MeOH. The solution was filtered through a 0.45 μm PTFE filter into a screw cap vial prior to injection into the HPLC system. The mobile phase used for the HPLC analysis was a gradient elution program consisting of UPW (A) and MeCN (B). The gradient was as follows: 10–95% B over 0–18 min, 95% B from 18–24 min, and 95–10% B from 25–30 min. The absorbance of the eluent was monitored at 210 nm. A 3 mL sample was introduced into the system at 30 °C and a flow rate of 4.7 mL/min was used, resulting in the isolation of 36.5 mg of the component of interest. This component was dissolved in 10 mL MeCN and UPW (80:20) and 1 mL was injected into a recycling HPLC system. The separation was performed with an isocratic elution of MeCN and UPW (80:20). The flow rate of the system was set at 4 mL/min and the absorbance was set to 210 nm. Thirty minutes were allotted for the column to condition and for the baseline to stabilize. After four complete cycles, 10 mg of compound **1** was eluted at minute 242. Each cycle took 60 min, and the entire cycle took 254 min to complete [22].

#### 3.4.3. Structural Elucidation

A microscope JM628 digital thermometer with an X-4 melting-point apparatus was used to determine the melting point. The ^1^H and ^13^C NMR spectra were acquired in deuterated methanol (MeOD) using a Bruker 600 Ultrashield NMR spectrometer at 600 and 150 MHz, respectively. UV and IR were measured using JASCO UV/Vis Spectrophotometer V-730 and Bruker FT-IR Spectrometer TENSOR II model, respectively.

## 4. Conclusions

The tandem LCMS-based phytochemicals analysis of the methanolic extract of *E. indica* through the integration of MZmine, GNPS, Compound Discoverer and SIRIUS platforms has managed to annotate and identify a total of 65 phytochemicals, comprising primary and secondary metabolites. It was found that all of these platforms complement each other by providing a wealth of information on the characterization, annotation, and identification of phytochemicals. The reliability of the technique was verified with the isolation of one of its consistently identified phytochemicals, **1**, known as loliolide. The structural elucidation of **1** using 1D and 2D NMR as well as comparison with literature values confirm that the isolated phytochemical is an anomer of loliolide at the C-3 position. This has consequently increased the level of confidence in the technique applied. The present work describes a tandem LCMS-based data analysis as a useful method for a fast and simultaneous identification of phytochemicals in *E. indica*, contributing to the study of the chemical properties of the genus and family. However, the elucidation of **1** as the anomer of the annotated compound also supports the limitation of MS spectra in differentiating stereoisomers. For this, additional experiments, including comparison with standards, are required to assign the absolute structure.

## Figures and Tables

**Figure 1 molecules-28-03111-f001:**
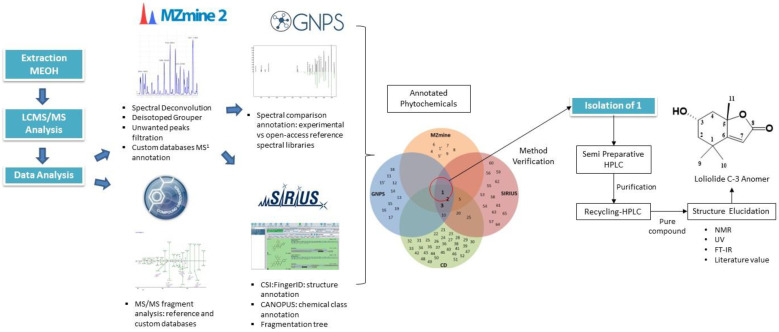
Infographic illustrating the annotation and identification process of phytochemicals from *E. indica* using high-performance liquid chromatography tandem mass spectrometry-based analysis.

**Figure 2 molecules-28-03111-f002:**
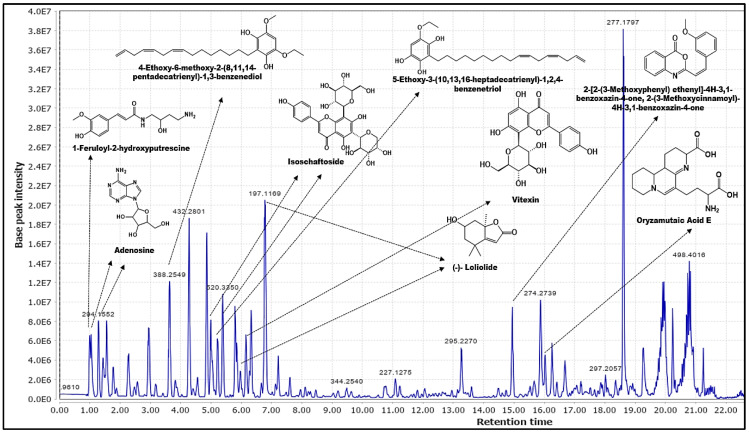
Identified phytochemicals from MZmine 2.53 based on TIC.

**Figure 3 molecules-28-03111-f003:**
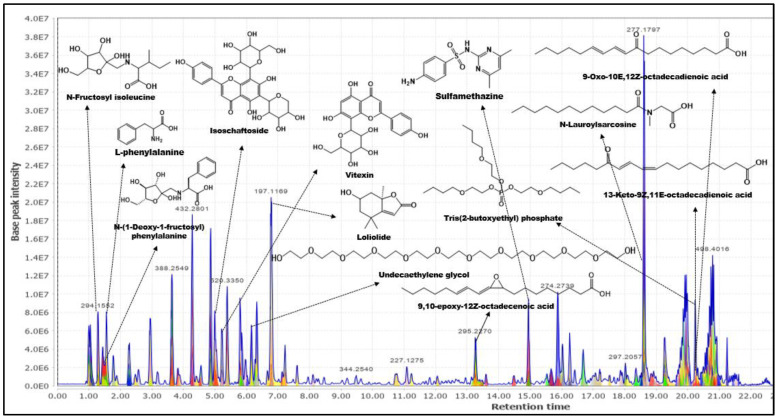
Annotated metabolites from GNPS based on TIC.

**Figure 4 molecules-28-03111-f004:**
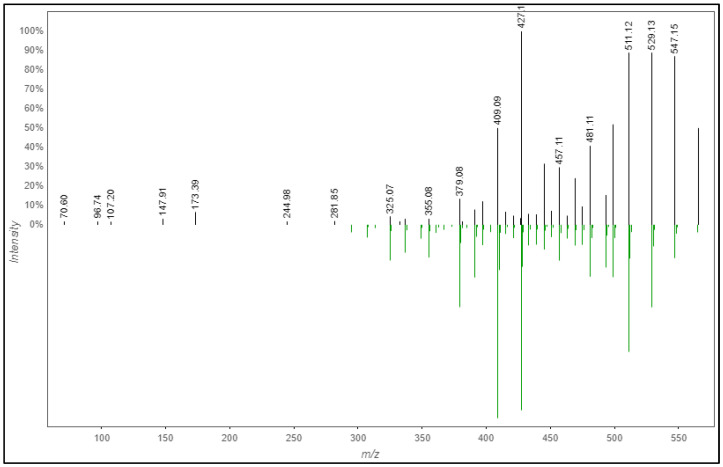
Mirror match of **2** with isoschaftoside with the reference spectra with gold library class and 0.82 cosine value.

**Figure 5 molecules-28-03111-f005:**
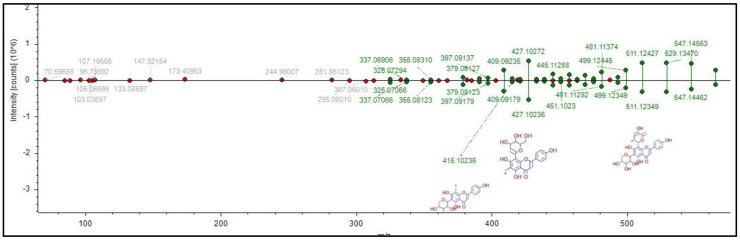
Mirror match of fragmentation of **2** with isoschaftoside by mzCloud spectral library.

**Figure 6 molecules-28-03111-f006:**
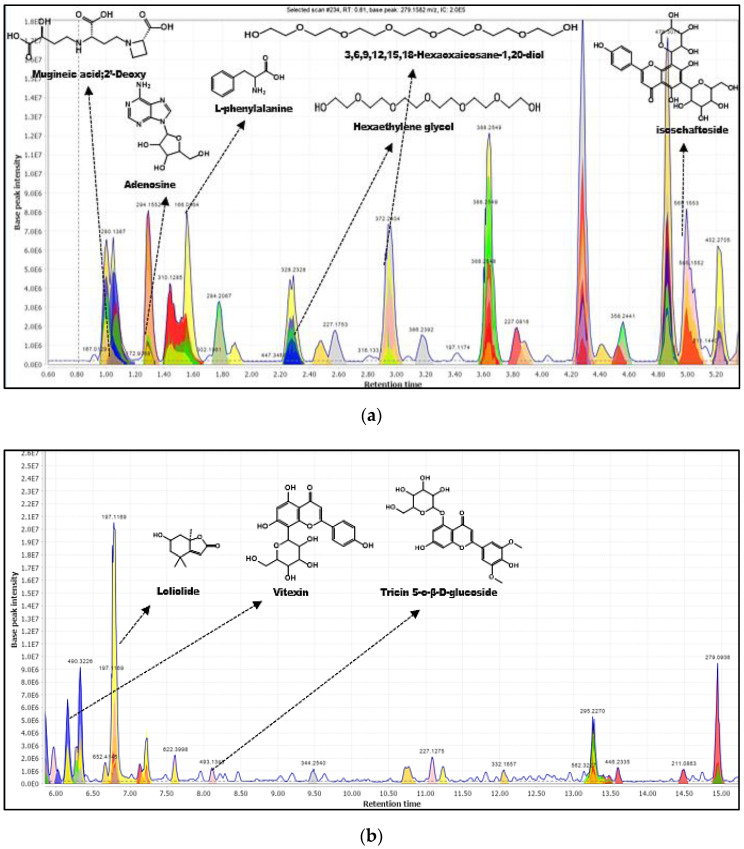
Annotated metabolites from CD based on TIC chromatogram. (**a**) retention time 0–6 min, (**b**) retention time 6–15 min, (**c**) retention time-13–22 min.

**Figure 7 molecules-28-03111-f007:**
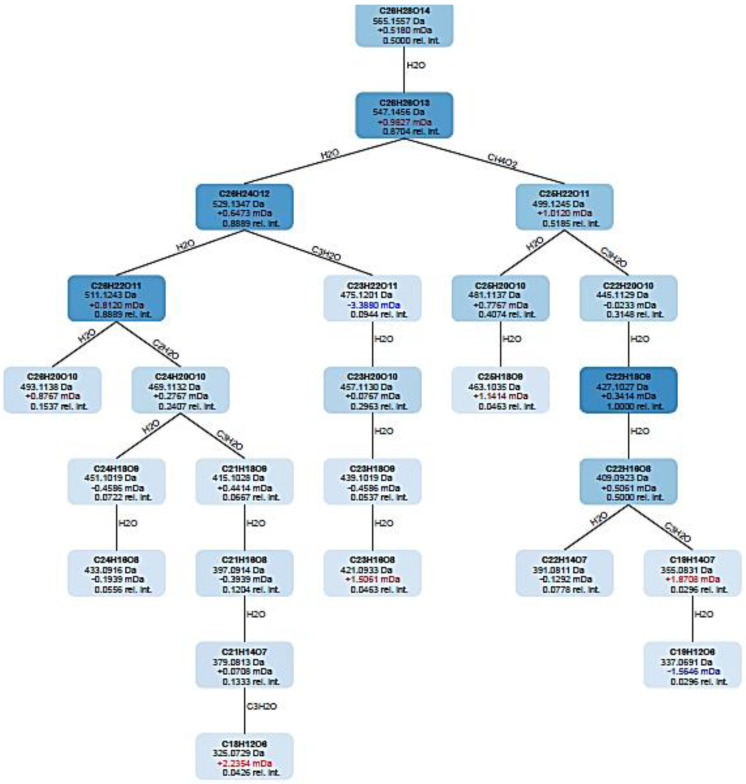
The fragmentation tree of candidate **2** with 565.16 *m/z* annotated as isoschaftoside.

**Figure 8 molecules-28-03111-f008:**
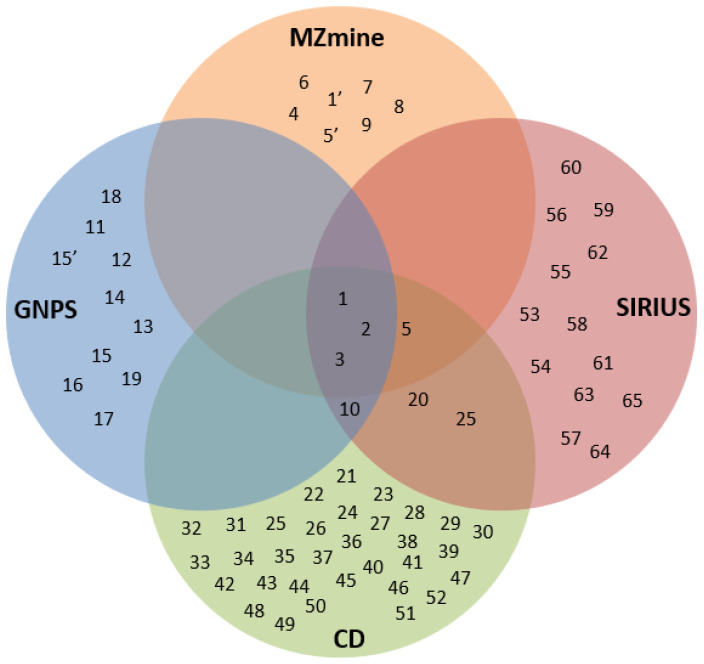
Venn diagram of all metabolites identified from *E. indica*. **1**, **1′**, Loliolide; **2**, **2′**, Isoschaftoside; **3**, vitexin; **4**, 4-Ethoxy-6-methoxy-2-(8,11,14-pentadecatrienyl)-1,3-benzenediol; **5**, **5′**, Adenosine; **6**, 5-Ethoxy-3-(10,13,16-heptadecatrienyl)-1,2,4-benzenetriol; **7**, Oryzamutaic Acid E; **8**, 1-Feruloyl-2-hydroxyputrescine; **9**, 2-(3-Methoxycinnamoyl)-4H-3,1-benzoxazin-4-one; **10**, L-phenylalanine; **11**, Tris(2-butoxyethyl) phosphate; **12**, Sulfamethazine; **13**, Undecaethylene glycol; **14**, 13-Keto-9*Z*,11*E*-octadecadienoic acid; **15**, **15′**, N-Fructosyl isoleucine; **16**, 9-Oxo-10*E*,12*Z*-octadecadienoic acid; **17**, N-(1-Deoxy-1-fructosyl) phenylalanine; **18**, N-Lauroylsarcosine; **19**, 9,10-epoxy-12Z-octadecenoic acid; **20**, 6-Gingerol; **21**, 13*S*-hydroxy-9*Z*,11*E*,15*Z*-octadecatrienoic acid; **22**, 3,5-Di-*tert*-butyl-4-hydroxybenzaldehyde; **23**, 5-Hydroxy-2-(4-hydroxy-3-methoxyphenyl)-6-methoxy; **24**, 7-Aminooctadecanoic acid; **25**, 9-Oxo-10*E*, 12*E*-octadecadienoic acid; **26**, Alpha-linolenic acid; **27**, Benazol-P; **28**, *Bis*(4-ethylbenzylidene) sorbitol; **29**, Bis(methylbenzylidene) sorbitol; **30**, Citroflex A-4; **31**, Hexaconazole; **32**, Mugineic acid; **33**, N-Lauryl sarcosine; **34**, Tricin 5-o-β-D-glucoside; **35**, 2,4,8,11-Dodecatetranoic acid; **36**, Decaethylene glycol; **37**, 3,6,9,12,15,18-Hexaoxaicosane-1,20-diol; **38**, Octaethylene glycol; **39**, Tris(2-butxyethyl) phosphate; **40**, 3,6,9,12,15-Pentaoxapentacosan-1-ol; **41**, 3,6,9,12,15-Tetraoxadocosan-1-ol; **42**, 3-[Dodecyl(2-hydroxyethyl) amino]-1,2-propanediol; **43**, 3′-Geranyl-3,4,2′,4′-tetrahydroxy-6′-methoxydihydrochalcone; **44**, 6-(1,3-Benzodioxol-5-yl)-2-oxo-4-phenyl-1,2-dihydro-3-pyridinecarbonitrile; **45**, Dibutyl phthalate; **46**, Methyl 9,10-dihydroxystearate; **47**, Methyl 9H-b-carboline-3-carboxylate; **48**, Hexaethylene glycol; **49**, N,N-Bis(2-hydroxyethyl)dodecanamide; **50**, N-Lauryldiethanolamine; **51**, Nonaethylene glycol; **52**, Safingol; **53**, 2,6-Bis[2-[2-(2-aminoethoxy) ethoxy] ethylamino] pyridine; **54**, N-[6-[3,5-diamino-2-[3-amino-6-(aminomethyl)-4,5-dihydroxyoxan-2-yl] oxy-6-hydroxycyclohexyl] oxyhexyl]-4-oxopentanamide; **55**, Istamycin C1; **56**, Netilmicin; **57**, N-{1-[(5-carbamimidamido-1-hydroxypentan-2-yl)-C-hydroxycarbonimidoyl]-2-methylpropyl}-2-[(1-hydroxyethylidene)amino]-4-methylpentanimidic acid; **58**, Plakoridine A; **59**, Schaftoside; **60**, Broussonetine M1; **61**, Fructose phenylalanine; **62**, Justicidin B; **63**, Isoleucinopine; **64**, Watsonol B; **65**, Embelin.

**Figure 9 molecules-28-03111-f009:**
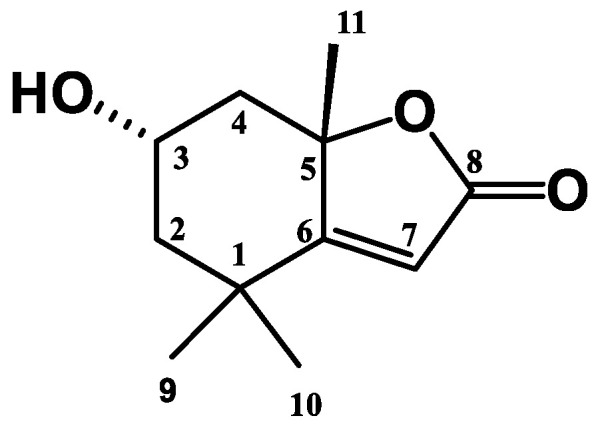
Structure of loliolide C-3 anomer.

**Table 1 molecules-28-03111-t001:** Phytochemicals hit from the Custom-Built Database imported to MZmine 2.53.

Compound Name	[M + H]^+^ (*m/z*)	Rt (min)	Database Sourced
Loliolide (**1**)	197.1174	6.78	LR_positive.csv
Isoschaftoside (**2**)	565.1553	5.00	LR_positive.csv
Vitexin (**3**)	433.1132	6.16	LR_positive.csv
4-Ethoxy-6-methoxy-2-(8,11,14-pentadecatrienyl)-1,3-benzenediol (**4**)	388.2603	3.63	DNP_Poaceae.csv
Adenosine (**5**)	268.1044	1.28	LR_positive.csv DNP_Poaceae.csv
Adenosine (**5′**)	268.1044	1.05	LR_positive.csv DNP_Poaceae.csv
5-Ethoxy-3-(10,13,16-heptadecatrienyl)-1,2,4-benzenetriol (**6**)	402.2763	5.22	DNP_Poaceae.csv
Loliolide (**1′**)	197.1174	5.97	LR_positive.csv
Oryzamutaic Acid E (**7**)	335.1830	16.02	DNP_Poaceae.csv
1-Feruloyl-2-hydroxyputrescine (**8**)	280.1425	1.04	DNP_Poaceae.csv
Isoschaftoside (**2′**)	565.1552	5.38	LR_positive.csv
2-(3-Methoxycinnamoyl)-4H-3,1-benzoxazin-4-one (**9**)	279.0906	14.95	DNP_Poaceae.csv

**Table 2 molecules-28-03111-t002:** Library hits from Feature-Based Molecular Networking in GNPS.

No	Compound Name	Cluster Index	Library Class	Cosine	MZ Error ppm	Spectral *m/z*/ Library *m/z*	Instrument	Ion Mode	Data Source	Ion Source
1	L-phenylalanine (**10**)	1	Bronze	1.00	0	166.09/ 166.09	Orbitrap	Positive	Trent Northen	LC-ESI
2	Vitexin (**3**)	770	Gold	0.96	5	433.11/ 433.11	Orbitrap	Positive	BMDMS-NP	ESI
3	Tris(2-butoxyethyl) phosphate (**11**)	660	Bronze	0.94	0	399.25/ 399.25	qTof	Positive	Nediljko Budisa	ESI
4	Loliolide (**1**)	25	Bronze	0.94	1	197.12/ 197.12	Orbitrap	Positive	Lihini Aluwihare	LC-ESI
5	Sulfamethazine (**12**)	222	Bronze	0.92	9	279.09/ 279.09	Hybrid FT	Positive	Massbank	ESI
6	Undecaethylene glycol (**13**)	928	Bronze	0.92	2	503.31/ 503.31	QQQ	Positive	Rob Knight	ESI
7	13-Keto-9*Z*,11*E*-octadecadienoic acid (**14**)	206	Bronze	0.85	1	277.22/ 277.22	qTof	Positive	Wolfender/Litaudon	ESI
8	N-Fructosyl isoleucine (**15**)	279	Bronze	0.84	4	294.15/ 294.15	qTof	Positive	Massbank	ESI
9	Isoshaftoside (**2**)	1053	Gold	0.82	0	565.16/ 565.16	Orbitrap	Positive	BMDMS-NP	ESI
10	9-Oxo-10*E*,12*Z*-octadecadienoic acid (**16**)	295	Bronze	0.82	2	295.23/ 295.23	IT-FT/ion trap with FTMS	Positive	Rob Knight	ESI
11	N-(1-Deoxy-1-fructosyl) phenylalanine (**17**)	448	Bronze	0.77	1	328.14/ 328.14	qTof	Positive	Claudia Maier	LC-ESI
12	N-Lauroylsarcosine (**18**)	175	Bronze	0.76	1	272.22/ 272.22	Hybrid FT	Positive	Massbank	ESI
13	N-Fructosyl isoleucine (**15′**)	188	Bronze	0.72	2	276.14/ 276.14	qTof	Positive	Massbank	ESI
14	9,10-epoxy-12*Z*-octadecenoic acid (**19**)	313	Bronze	0.72	2	297.24/ 297.24	HCD	Positive	Rob Knight	ESI

**Table 3 molecules-28-03111-t003:** Phytochemicals hits from the Compound Discoverer platform.

No	Compound Name	Molecular Formula	[M + H]^+^	Fragmentation Ions (MS/MS)	Rt (min)	∆Mass [Da]	∆Mass [ppm]	mzCloud Similarity	Reference	FISh Score	MIC *
1	6-Gingerol (**20**)	C_17_H_26_O_4_	295.1905	277.1803, 173.4033	16.029	0.00012	0.42	70.0	ChemSpider	85.3	2
2	-13*S*-hydroxy-9*Z*,11*E*,15*Z*-octadecatrienoic acid (**21**)	C_18_H_30_O_3_	295.2270	277.2160, 151.1113	13.268	0.00019	0.64	89.8	DNP, ChemSpider	89.71	2
3	3,5-Di-*tert*-butyl-4-hydroxybenzaldehyde (**22**)	C_15_H_22_O_2_	235.1692	219.1388, 179.1068	19.915	0.00007	0.30	95.9	DNP, Chemspider	80	2
4	5-Hydroxy-2-(4-hydroxy-3-methoxyphenyl)-6-methoxy (**23**)	C_23_H_24_O_12_	493.1350	331.08015, 313.7641	7.132	0.00038	0.77	99.3	ChemSpider	73.91	2
5	7-Aminooctadecanoic acid (**24**)	C_18_H_37_NO_3_	316.2849	106.4888, 115.0000	20.015	0.00051	1.60	-	DNP, ChemSpider	100.00	2
6	9-Oxo-10*E*, 12*E*-octadecadienoic acid (**25**)	C_18_H_30_O_3_	295.2268	277.1609, 151.1115	20.489	0.00013	0.46	90.6	ChemSpider, DNP	77.78	2
7	Adenosine (**5**)	C_10_H_13_N_5_O_4_	268.1041	136.0612, 112.7441	1.286	0.00028	1.04	99.3	DNP ChemSpider	83.33	2
8	Alpha-linolenic acid (**26**)	C_18_H_30_O_2_	279.2319	279.2311, 261.2196	20.898	0.00008	0.28	89.5	CD Database	74.66	2
9	Benazol-P (**27**)	C_13_H_11_N_3_O	226.0975	126.0975, 120.0550	20.731	0.00003	0.13	95.8	CD Database	29.41	2
10	*Bis*(4-ethylbenzylidene) sorbitol (**28**)	C_24_H_39_O_6_	415.2122	135.0799, 119.0851	17.867	0.00116	2.80	99.8	CD Database	75.00	2
11	*Bis*(methylbenzylidene) sorbitol (**29**)	C_22_H_26_O_6_	387.1802	119.0491, 105.0692	16.258	0.00056	1.46	99.8	CD Database	44.44	2
12	Citroflex A-4 (**30**)	C_20_H_34_O_8_	403.2331	185.0806, 129.0175	21.247	0.00046	1.46	95.3	CD Database	54.55	2
13	Isoschaftoside (**2**)	C_26_H_28_O_14_	565.1564	427.1027, 409.0923	4.996	0.00113	2.00	97.1	DNP	45.16	2
14	Hexaconazole (**31**)	C_14_H_17_C_12_N_3_O	314.0824	184.9910, 158.9758	18.355	0.00056	1.78	98.1	ChemSpider	70.00	2
15	L-phenylalanine (**10**)	C_9_H_11_NO_2_	166.0861	120.0802, 103.0537	1.557	0.00003	−0.17	99.9	ChemSpider	87.50	2
16	Mugineic acid (**32**)	C_12_H_20_N_2_O_7_	305.1344	227.1023, 191.0811	1.023	0.00035	1.14	90	DNP	71.43	2
17	N-Lauryl sarcosine (**33**)	C_15_H_29_NO_3_	272.2946	230.4062, 90.0543	18.662	0.00018	0.66	77.8	ChemSpider	82.35	2
18	Tricin 5-o-β-D-glucoside (**34**)	C_23_H_24_O_12_	493.1350	331.0818, 329.4911	8.108	0.00038	0.77	99.2	ChemSpider	66.67	2
19	Vitexin (**3**)	C_21_H_20_O_10_	433.1136	397.0921, 313.0708	6.159	0.00051	1.18	97.8	ChemSpider	22.58	2
20	2,4,8,11-Dodecatetranoic acid (**35**)	C_16_H_25_NO	248.2011	117.0693, 105.0693	20.684	0.00001	0.03	-	DNP	71.43	3
21	Decaethylene glycol (**36**)	C_20_H_42_O_11_	459.2809	177.1119, 133.0855	4.864	0.00071	1.54	88	ChemSpider	85.71	2
22	3,6,9,12,15,18-Hexaoxaicosane-1,20-diol (**37**)	C_14_H_30_O_8_	327.2018	173.4045, 146.7485	2.949	0.00068	2.09	92.2	CD Database	75.00	2
23	Octaethylene glycol (**38**)	C_16_H_34_O_9_	371.2282	329.1156, 133.0853,	3.634	0.00050	1.34	97.2	ChemSpider	100	2
24	Tris(2-butxyethyl) phosphate (**39**)	C_18_H_39_O_7_P	399.2512	299.1618, 199.0726	20.240	0.00067	2.68	98.0	CD Database	77.8	3
25	3,6,9,12,15-Pentaoxapentacosan-1-ol (**40**)	C_20_H_42_O_6_	379.3095	173.3948, 89.0958	20.021	0.00056	1.48	84.6	CD Database	82.4	4
26	3,6,9,12,15-Tetraoxadocosan-1-ol (**41**)	C_18_H_38_O_5_	335.2802	133.0855, 89.059245	20.053	0.00046	1.37	-	CD Database	83.33	2
27	3-[Dodecyl(2-hydroxyethyl) amino]-1,2-propanediol (**42**)	C_17_H_37_NO_3_	304.2850	256.2627, 122.0806	17.006	0.00041	1.36	-	ChemSpider	80.00	3
28	3′-Geranyl-3,4,2′,4′-tetrahydroxy-6′-methoxydihydrochalcone (**43**)	C_26_H_32_O_6_	441.2253	173.4048, 159.6250	18.919	−0.00185	−4.21	-	Arita Lab 6549 Flavonoid S	100.00	2
29	Loliolide (**1**)	C_11_H_16_O_3_	197.1173	179.1063, 133.1007	6.781	0.00007	0.37	-	DNP	46.15	3
30	6-(1,3-Benzodioxol-5-yl)-2-oxo-4-phenyl-1,2-dihydro-3-pyridinecarbonitrile (**44**)	C_19_H_12_N_2_O_3_	317.0925	289.0975, 271.0869	17.520	0.00049	1.55	-	ChemSpider	44.83	3
31	Dibutyl phthalate (**45**)	C_16_H_22_O_4_	279.1592	233.0775, 149.0229	20.694	0.00012	0.45	86	CD Database	24.24	2
32	Methyl 9,10-dihydroxystearate (**46**)	C_19_H_38_0_4_	331.2845	285.2999, 173.3954	19.266	0.00040	1.21	82	CD Database	100.00	3
33	Methyl 9H-b-carboline-3-carboxylate (**47**)	C_13_H_10_N_2_O_2_	227.0814	210.0726, 182.0786	3.826	−0.00002	−0.10	73	ChemSpider	-	3
34	Hexaethylene glycol (**48**)	C_12_H_26_O_7_	283.1752	270.8245, 173.3921	2.277	0.00010	0.34	80	ChemSpider	100.00	3
35	N,N-Bis(2-hydroxyethyl)dodecanamide (**49**)	C_16_H_33_NO_3_	288.2534	228.2041, 106.0856	17.915	0.00031	1.08	78	CD Database	50.00	2
36	N-Lauryldiethanolamine (**50**)	C_16_H_35_NO_2_	274.2741	106.0862, 88.0751	15.873	0.00024	0.86	-	CD Database	70.00	2
37	Nonaethylene glycol (**51**)	C_18_H_38_O_10_	415.2544	221.1385, 177.1166	4.279	0.00064	1.54	83	ChemSpider	100.00	2
38	Safingol (**52**)	C_18_H_39_NO_2_	302.3055	102.2445	18.091	0.00009	0.31	93	ChemSpider	66.67	2

* Metabolite Identification Confidence (MIC); Level 1: unambiguous identification by comparison of the retention time and MS/MS fragmentation with reference standards; Level 2: putative identification through MS/MS fragmentation libraries without the presence of standards; Level 3: tentative structure determination by matching MS^1^ information with the compound database; Level 4: matching with the molecular formula, isotope abundance distribution, adduct ion determination charge state, and ion determination; and Level 5: annotation through unique features such as mass measurement accuracy (±ppm).

**Table 4 molecules-28-03111-t004:** Identified phytochemicals from SIRIUS 4.0 Software.

No	Annotation (CSI: FingerID)	[M + H]^+^	Fragment Ions	Rt (min)	Molecular Formula	COSMIC	Classification	Smiles	Links	MIC *
1	2,6-Bis[2-[2-(2-aminoethoxy) ethoxy] ethylamino] pyridine (**53**)	372.64	327.2019, 133.0853	2.94	C_17_H_33_N_5_O_4_	0.8221	Aminopyrimidines and derivatives	C1=CC(=NC(=C1)NCCOCCOCCN)NCCOCCOCCN	PubChem:(86235228)	2
2	N-[6-[3,5-diamino-2-[3-amino-6-(aminomethyl)-4,5-dihydroxyoxan-2-yl] oxy-6-hydroxycyclohexyl] oxyhexyl]-4-oxopentanamide (**54**)	520.33	503.3067, 133.0851	5.39	C_23_H_45_N_5_O_8_	0.722	Dialkyl ether	CC(=O)CCC(=O)NCCCCCCOC1C(C(CC(C1OC2C(C(C(C(O2)CN)O)O)N)N)N)O	PubChem:(90164161)	2
3	Istamycin C1 (**55**)	432.28	415.2545, 133.0854	4.28	C_19_H_37_N_5_O_6_	0.611	alkanolamine	CCNCC1CCC(C(O1)OC2C(CC(C(C2O)N(C)C(=O)CNC=O)OC)N)N	COCONUT, KEGG, Natural products, CHEBI, PubChem	2
4	Netilmicin (**56**)	476.31	459.2808, 133.0854	4.87	C_21_H_41_N_5_O_7_	0.595	Dialkyl ether	CCNC1CC(C(C(C1OC2C(C(C(CO2)(C)O)NC)O)O)OC3C(CC=C(O3)CN)N)N	KEGG, COCONUT, Natural	2
5	Adenosine (**5**)	268.10	136.0612	1.28	C_10_H_13_N_5_O4	0.399	Purine nucleosides	C1=NC(=C2C(=N1)N(C=N2)C3C(C(C(O3)CO)O)O)N	HMDB, SuperNatural HSDB, MeSH, Plantcyc, PubMed, NORMAN, COCONUT, KNApSAcK, Natural Products, PubChem, CHEBI, KEGG	2
6	N-{1-[(5-carbamimidamido-1-hydroxypentan-2-yl)-C-hydroxycarbonimidoyl]-2-methylpropyl-}2-[(1-hydroxyethylidene) amino]-4-methylpentanimidic acid (**57**)	415.30	119.0851, 145.0637	20.87	C_19_H_38_N_6_O_4_	0.371	Dipeptides	CC(C)CC(C(=O)NC(C(C)C)C(=O)NC(CCCN=C(N)N)CO)NC(=O)C	COCONUT:(CNP0335012)	2
7	Plakoridine A (**58**)	572.44	555.4106, 133.0851	19.85	C_35_H_57_NO_5_	0.368	Fatty amides	CCCCCCCCCCCCCCCCC(=O)C=C1C(C(C(N1CCC2=CC=C(C=C2)O)CCC)C(=O)OC)O	Natural product, COCONUT	2
8	Isoschaftoside (**2**)	565.16	547.1456, 427.1027	5.00	C_26_H_28_O_14_	0.350	Flavanoid 8-C- Glycoside	C1C(C(C(C(O1)C2=C3C(=C(C(=C2O)C4C(C(C(C(O4)CO)O)O)O)O)C(=O)C=C(O3)C5=CC=C(C=C5)O)O)O)O	CHEBI, COCONUT, HMDB, KNApSAcK, MeSH, Natural Products, Pubchem, PubMed, SuperNatural, ZINC bio	2
9	Schaftoside (**59**)	565.17	529.1347, 427.1027	5.36	C_26_H_28_O_14_	0.350	Flavanoid 8-C- Glycoside	C1C(C(C(C(O1)C2=C(C(=C3C(=C2O)C(=O)C=C(O3)C4=CC=C(C=C4)O)C5C(C(C(C(O5)CO)O)O)O)O)O)O)O	CHEBI, COCONUT, HMDB, KEGG, KNApSAcK, MeSH, Natural Products, Pubchem, PubMed, SuperNatural, ZINC bio	2
10	Broussonetine M1(**60**)	348.28	173.1167, 155.1062	13.26	C_18_H_37_NO_5_	0.295	Long chain fatty acid	C(CCCCC(CCCCO)O)CCCC1C(C(C(N1)CO)O)O	COCONUT, Natural Product, PubChem, PubMed, KNapSAck, Supernatural	2
11	Vitexin (**3**)	433.11	397.0921, 313.0708	6.11	C_21_H_20_O_10_	0.283	Phenolic glycoside	C1=CC(=CC=C1C2=CC(=O)C3=C(O2)C(=C(C=C3O)O)C4C(C(C(C(O4)CO)O)O)O)O	HMDB, SuperNatural, ZINC, MeSH, Plantcyc, PubMed, NORMAN, COCONUT, KNApSAcK, Natural Products, CHEBI, PubChem, KEGG, Biocyc	2
12	Fructose phenylalanine (**61**)	328.14	292.118, 264.1228	1.45	C_15_H_21_NO_7_	0.219	Hexoses	C1=CC=C(C=C1)CC(C(=O)O)NCC(=O)C(C(C(CO)O)O)O	MeSH, PubMed	2
13	Loliolide (**1**)	197.12	179.1063, 133.1007	6.78	C_11_H_16_O_3_	0.171	Dihydrofurans	O=C1OC2(C(=C1)C(C)(C)CC(O)C2)C	CHEBI, COCONUT, PubChem, PubMed, Natural Product, MeSH, KNapSAck, Supernatural	2
14	Justicidin B (**62**)	365.11	203.0527, 185.0419	1.00	C_21_H_16_O_6_	0.158	Phenolic Glycoside	COC1=CC2=CC3=C(C(=C2C=C1OC)C4=CC5=C(C=C4)OCO5)C(=O)OC3	CHEBI, COCONUT, PubChem, PubMed, Natural Product, MeSH, KNapSAck, Supernatural	2
15	Isoleucinopine (**63**)	262.13	244.1180, 216.1230	1.05	C_11_H_19_NO_6_	0.136	Alpha Amino Acid	O=C(O)CCC(NC(C(=O)O)C(C)CC)C(=O)O	COCONUT, Natural Product, Pubchem	2
16	Gingerol (**20**)	295.19	277.1803, 137.0592	16.03	C_17_H_26_O_4_	0.093	Carboxylic acids	O=C(CCC1=CC=C(O)C(OC)=C1)CC(O)CCCCC	CHEBI, COCONUT, PubChem, PubMed, Natural Product, MeSH, KNapSAck, Supernatural	2
17	11-Oxooctadeca-9,12-dienoic acid (**25**)	295.23	277.2160, 259.2054	13.26	C_18_H_30_O_3_	0.071	Lineolic acids and derivatives	O=C(C=CCCCCC)C=CCCCCCCCC(=O)O	COCONUT, PubChem, Natural Product, KNapSAck, Supernatural	2
18	L-phenylalanine (**10**)	166.09	131.0486, 120.0802	1.55	C_9_H_11_NO_2_	0.736	Phenylalanine and derivatives	O=C(O)C(N)CC=1C=CC=CC1	CHEBI, COCONUT, PubChem, PubMed, Natural Product, MeSH, KNapSAck, Supernatural, NORMAN, HMDB, Plantcyc	2
19	Watsonol B (**64**)	277.18	231.1739, 137.0591	18.61	C_17_H_24_O_3_	0.109	Prenol lipids	O=C(OCC(=C)C1CCC(=C)C2CCC(=C)C2C1O)C	COCONUT, KNapSAcK, Natural Products, PubChem, SuperNatural	2
20	Embelin (**65**)	295.19	203.1792, 137.0592f	18.61	C_17_H_26_O_4_	0.239	Prenol lipids	CCCCCCCCCCCC1=C(C(=O)C=C(C1=O)O)O	COCONUT, KNapSAcK, Natural Products, PubChem, SuperNatural, CHEBI, KEGG, MeSH, ZINC bio, NORMAN	2

* Metabolite Identification Confidence (MIC); Level 1: unambiguous identification by comparison of the retention time and MS/MS fragmentation with reference standards; Level 2: putative identification through MS/MS fragmentation libraries without the presence of standards; Level 3: tentative structure determination by matching MS^1^ information with the compound database; Level 4: matching with the molecular formula, isotope abundance distribution, adduct ion determination charge state, and ion determination; and Level 5: annotation through unique features such as mass measurement accuracy (±ppm).

**Table 5 molecules-28-03111-t005:** List of characterized phytochemicals from SIRIUS software.

No	[M + H]^+^	RT (min)	Molecular Formula	Class of Compound	COSMIC
1	498.40	20.79	C_26_H_51_N_5_O_4_	Aminopyrimidines and derivatives	0.730
2	454.38	20.82	C_24_H_47_N_5_O_3_	Azoles	0.310
3	542.43	20.73	C_28_H_55_N_5_O_5_	Amino acids and derivatives	0.378
4	388.25	3.64	C_17_H_33_N_5_O_5_	Heteroaromatic Compounds	0.607
5	484.38	19.95	C_25_H_49_N_5_O_4_	Aminopyrimidines and derivatives	0.735
6	528.41	19.91	C_27_H_53_N_5_O_5_	Benzenoids	0.546
7	274.27	15.88	C_16_H_35_NO_2_	1,2-aminoalcohols	0.087
8	440.36	19.99	C_23_H_45_N_5_O_3_	Heteroaromatic Compounds	0.652
9	279.09	14.95	C_12_H_11_FN_4_O_3_	Benzoyl derivatives	0.314
10	399.25	20.23	C_18_H_39_O_7_P	Trialkyl phosphates	0.190
11	490.32	6.33	C_28_H_43_NO_6_	Benzenoids	0.293
12	586.45	20.69	C_30_H_59_N_5_O_6_	Peptides	0.372
13	572.24	19.85	C_29_H_57_N_5_O_6_	N-Acyl amines	0.370
14	294.16	1.29	C_12_H_23_NO_7_	Alpha Amino Acids and derivatives	0.111
15	415.25	4.28	C_18_H_38_O_10_	Polyethylene glycols	0.303
16	454.38	20.82	C_24_H_47_N_5_O_3_	Azoles	0.310
17	327.22	18.63	C_18_H_30_O_5_	Fatty acid methyl esters	0.063
18	372.26	2.94	C_17_H_33_N_5_O_4_	Aminopyrimidines and derivatives	0.855
19	276.14	1.28	C_8_H_17_N_7_O_4_	Primary amines	0.096
20	371.23	3.64	C_16_H_34_O_9_	Polyethylene glycols	0.329
21	534.35	6.80	C_24_H_47_N_5_O_8_	Dialkyl ethers	0.138
22	484.39	19.95	C_25_H_49_N_5_O_4_	Aminopyrimidines and derivatives	0.735
23	280.14	1.05	C_11_H_21_NO_7_	Alpha amino acids	0.170
24	520.33	5.39	C_23_H_45_N_5_O_8_	Dialkyl ethers	0.632
25	402.27	5.21	C_18_H_35_N_5_O_5_	Dialkyl ethers	0.474
26	459.28	4.87	C_20_H_42_O_11_	Polyethylene glycols	0.298
27	446.29	5.79	C_20_H_39_N_5_O_6_	Heteroaromatic Compounds	0.102
28	388.25	3.64	C_17_H_33_N_5_O_5_	Heteroaromatic Compounds	0.606
29	344.23	2.95	C_15_H_29_N_5_O_4_	Dialkyl ethers	0.625
30	564.34	5.85	C_25_H_49_N_5_O_9_	Carboxylic acid amides	0.103
31	387.18	16.25	C_29_H_20_	Phenols	0.219
32	630.48	20.64	C_23_H_63_N_5_O_7_	Fatty amides	0.372
33	410.35	20.85	C_22_H_43_N_5_0_2_	Aminopyrimidines and derivatives	0.193
34	388.26	3.63	C_17_H_33_N_5_O_5_	Aryl thiothers	0.605
35	304.30	19.27	C_21_H_37_N	Phenylmethylamines	0.278
36	440.36	19.99	C_23_H_45_N_5_O_3_	Heteroaromatic compounds	0.463
37	268.10	1.05	C_15_H_13_BO_4_	Glycosyl compounds	0.506
38	328.23	2.29	C_15_H_29_N_5_O_3_	Aminopyrimidines and derivatives	0.416
39	277.18	16.03	C_17_H_24_O_3_	Prenol lipids	0.135
40	371.23	3.64	C_16_H_34_O_9_	Polyethylene glycol	0.329
41	310.13	1.44	C_15_H_19_NO_6_	Phenylalanine and derivatives	0.145
42	372.26	2.94	C_17_H_33_N_5_O_4_	Aminopyrimidines and derivatives	0.855

**Table 6 molecules-28-03111-t006:** ^1^H- (in CD_3_OD, 600 MHz) and ^13^C- APT NMR (in CD_3_OD, 150 MHz) data of compound **1** as loliolide and ^a^ comparison with literature data from Yuan et al., 2018 [36].

Loliolide								
Position	^δ^_H_ (ppm)	^δ^_C_ (ppm)	^δ^_H_ (ppm) ^a^ (CD_3_OD)	^δ^_C_ (ppm) ^a^ (CD_3_OD)	HMBC	HMQC	COSY	NOESY
1	-	35.73	-	35.75	-	-	-	-
2	α: 2.01 (*dt*, *J* = 14.4, 3.1) β: 1.55 (*dd*, *J* = 14.4, 3.7,)	49.53	α: 1.50 (*dt*, *J* = 14.4, 3.1) β: 1.98 (*dd*, *J* = 4.0, 16.0)	46.53	C-1, C-4, C-3, C-8 C-1, C-10, C-11	C-2	H2β, H3 H2α, H3	-
3	4.33 (*m*)	66.1	4.19	65.80	C-1, C-5, C-11	C-3	H2α, H11	H11, H2β, H9
4	α: 1.78 (*d*, *J* = 4.0) β: 2.44 (*dt*, *J* = 13.6, 2.4)	46.99	α: 1.71 (*d*, *J* = 4.0) β: 2.46 (*dt*, *J* = 4.0, 16.0)	44.99	C-3, C-4, C-5, C-8, C-10 C-3, C-4, C-5, C-8, C-9	C-4	H4β, H3 H2α, H3	-
5	-	87.8	-	87.52	-	-	-	-
6	-	170.1	-	172.12	-	-	-	-
7	5.69 (*s*)	112.2	5.80	111.88	C-1, C-5, C-6, C-8, C-9	C-7	-	-
8	-	181.5	-	184.13	-	-	-	-
9	1.47	25.8	1.45	25.52	C-1, C-4, C-8, C-11	C-9	-	-
10	1.27	26.3	1.26	29.57	C-1, C-3, C-4, C-5, C-7, C-8, C-9	C-10	-	-
11	1.78	29.8	1.74	25.99	C-3, C-4, C-5, C-8, C-10	C-11	-	-

## Data Availability

The data presented in this study are available on request from corresponding author. The data are not publicly available due to being part of a research project.

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
