# Peer review of "Annotation and Identification of Phytochemicals from Eleusine indica Using High-Performance Liquid Chromatography Tandem Mass Spectrometry: Databases-Driven Approach"

_molecules, 2023, doi:10.3390/molecules28073111_

Round 1
Reviewer 1 Report
Dear esteemed authors,
Having undertaken a meticulous assessment of your paper, I find the content to be captivating. Nonetheless, there are certain points that require your attention to further enhance the quality of the paper. I would highly appreciate it if you could thoroughly address my comments and furnish any necessary supplementary materials. Additionally, I kindly request that the revised sections be clearly highlighted in the main manuscript file.
Thank you for your valuable contributions to the field. Please check the following comments and provide your answers at the due time.
1- Please provide a flowchart to summarize the M&M section.
2- Please add the botanical figure of the studied species.
3- The introduction section is too long. Please summarize it into three paragraphs with a maximum of 10-12 references. Please make sure all cited literature has validated DOI identifiers.
4- There are some grammatical errors inside the manuscript text. Please carefully revise the manuscript content and try to use active language in discussing the results.
5- Please highlight the novelty of this study at the end of the introduction section
6- Please make sure all applied protocols and scientific procedures were cited correctly.
7- Many figures have been used within the paper. Please supplement some of these figures and try to add only the necessary figures to the paper. Figures’ captions also require elaboration and the respected authors should add enough explanations to each figure caption.
8- In some cases, Figures have low quality. The respected authors inserted the screenshot of target platforms used for metabolic annotation of the studied plant. Please rearrange the figures, extract essential data and plots from each screenshot, and add only the necessary information to the paper. This helps academic readers of your paper to easily understand the content.
9- Please use a Venn diagram or circular stacked bar plot to compare the content of identified metabolites based on each detection system used in this study.
10- Which metabolites were abundant in this plant and were identified commonly in each platform?
11- Please remove all typographical errors in the figure's inner labels. The beauty of figures is a pivotal infrastructure of their scientific content.
12- The paper's content is novel but the way of representation is weak. In some sections, confusing sentences have been written which makes the section difficult to understand. Please thoroughly revise the manuscript content and restructure your explanations.
13- Please use the abbreviated style for journals in the reference list.
14- Based on the authors' results, which studied platform can be recommended for further studies in this field? Please summarize the weakness of each platform in detecting and identifying secondary metabolites.
Author Response
|
Reviewers
|
Comment and suggestion |
Response |
|
1
|
Dear esteemed authors,
Having undertaken a meticulous assessment of your paper, I find the content to be captivating. Nonetheless, there are certain points that require your attention to further enhance the quality of the paper. I would highly appreciate it if you could thoroughly address my comments and furnish any necessary supplementary materials. Additionally, I kindly request that the revised sections be clearly highlighted in the main manuscript file.
Thank you for your valuable contributions to the field. Please check the following comments and provide your answers at the due time. |
Thank you
All the revised content based on your comments and suggestions are highlighted in RED. |
|
1- Please provide a flowchart to summarize the M&M section. |
Thank you for the suggestion. We decided to come out with an infographic summarizing the content. This is assigned as Figure 1 in the revised manuscript. |
|
|
2- Please add the botanical figure of the studied species. |
Thank you for the suggestion. However, we decided not to include the botanical figure of the studied species as it is not necessary in the content of our research. |
|
|
3- The introduction section is too long. Please summarize it into three paragraphs with a maximum of 10-12 references. Please make sure all cited literature has validated DOI identifiers. |
The introduction section has been revised accordingly. |
|
|
4- There are some grammatical errors inside the manuscript text. Please carefully revise the manuscript content and try to use active language in discussing the results. |
The grammatical errors have been corrected. |
|
|
5- Please highlight the novelty of this study at the end of the introduction section |
The novelty of the study has been now included in the last paragraph of the introduction section. |
|
|
6- Please make sure all applied protocols and scientific procedures were cited correctly. |
Citation has been included in the protocol section where applicable. |
|
|
7- Many figures have been used within the paper. Please supplement some of these figures and try to add only the necessary figures to the paper. Figures’ captions also require elaboration and the respected authors should add enough explanations to each figure caption. |
Some of the figures have been removed. Only figures that necessary are left in the revised version. |
|
|
8- In some cases, Figures have low quality. The respected authors inserted the screenshot of target platforms used for metabolic annotation of the studied plant. Please rearrange the figures, extract essential data and plots from each screenshot, and add only the necessary information to the paper. This helps academic readers of your paper to easily understand the content. |
The low-quality figures have been replaced accordingly. |
|
|
9- Please use a Venn diagram or circular stacked bar plot to compare the content of identified metabolites based on each detection system used in this study. |
Venn Diagram has been used now to compare the phytochemicals in each platform used. Thank you for your suggestion. |
|
|
10- Which metabolites were abundant in this plant and were identified commonly in each platform? |
We did not do quantification work here. Some peaks give the high intensity of absorption in the TIC, however, it is not fair to judge from this as different compounds have different total ion abundance, which is based on the stability of ions produced. Anyway, this is good to consider in the future. |
|
|
11- Please remove all typographical errors in the figure's inner labels. The beauty of figures is a pivotal infrastructure of their scientific content. |
This has been removed. |
|
|
12- The paper's content is novel but the way of representation is weak.
In some sections, confusing sentences have been written which makes the section difficult to understand. Please thoroughly revise the manuscript content and restructure your explanations. |
The structure of the sentences and the content of the manuscript has been revised accordingly. |
|
|
13- Please use the abbreviated style for journals in the reference list. |
Endnote has been applied to follow the journal referencing format. |
|
|
14- Based on the authors' results, which studied platform can be recommended for further studies in this field? Please summarize the weakness of each platform in detecting and identifying secondary metabolites. |
This has been included in section 2.2 |

Reviewer 2 Report
The work is devoted to the phytochemical characterization of Eleusine indica using the LC-MS technique. A distinctive feature of the work is the combination of various databases for the identification of plant metabolites. The work is well planned and described. The authors probably should not have described the process of identifying the components of Eleusine indica in such detail, providing screenshots of the working windows of various programs, which led to a significant amount of the manuscript.
Author Response
|
Reviewers
|
Comment and suggestion |
Response |
|
2 |
The work is devoted to the phytochemical characterization of Eleusine indica using the LC-MS technique. A distinctive feature of the work is the combination of various databases for the identification of plant metabolites.
The work is well planned and described.
The authors probably should not have described the process of identifying the components of Eleusine indica in such detail, providing screenshots of the working windows of various programs, which led to a significant amount of the manuscript. |
Thank you.
The content has been revised accordingly. |

Round 2
Reviewer 1 Report
Suitable for publication.